# Exploring the Potential of Malvidin and Echiodinin as Probable Antileishmanial Agents Through In Silico Analysis and In Vitro Efficacy

**DOI:** 10.3390/molecules30010173

**Published:** 2025-01-04

**Authors:** Luis Daniel Goyzueta-Mamani, Daniela Pagliara Lage, Haruna Luz Barazorda-Ccahuana, Margot Paco-Chipana, Mayron Antonio Candia-Puma, Gonzalo Davila-Del-Carpio, Alexsandro Sobreira Galdino, Ricardo Andrez Machado-de-Avila, Rodolfo Cordeiro Giunchetti, Edward L. D’Antonio, Eduardo Antonio Ferraz Coelho, Miguel Angel Chávez-Fumagalli

**Affiliations:** 1Computational Biology and Chemistry Research Group, Vicerrectorado de Investigación, Universidad Católica de Santa María, Arequipa 04000, Peru; lgoyzueta@ucsm.edu.pe (L.D.G.-M.); hbarazorda@ucsm.edu.pe (H.L.B.-C.); 74252022@ucsm.edu.pe (M.P.-C.); mcandia@ucsm.edu.pe (M.A.C.-P.); 2Programa de Pós-Graduação em Ciências da Saúde: Infectologia e Medicina Tropical, Faculdade de Medicina, Universidade Federal de Minas Gerais, Belo Horizonte 31270-901, Brazil; dpagliarara@icb.ufmg.br (D.P.L.); eduardoferrazcoelho@yahoo.com.br (E.A.F.C.); 3Facultad de Ciencias Farmacéuticas, Bioquímicas y Biotecnológicas, Universidad Católica de Santa María, Arequipa 04000, Peru; gdavilad@ucsm.edu.pe; 4Laboratório de Biotecnologia de Microrganismos, Universidade Federal São João Del-Rei, Divinópolis 35501-296, Brazil; asgaldino@ufsj.edu.br; 5Instituto Nacional de Ciência e Tecnologia em Biotecnologia Industrial (INCT-BI), Distrito Federal, Brasilia 70070-010, Brazil; 6Programa de Pós-Graduação em Ciências da Saúde, Universidade do Extremo Sul Catarinense, Criciúma 88806-000, Brazil; r_andrez@unesc.net; 7Laboratório de Biologia das Interações Celulares, Instituto de Ciências Biológicas, Universidade Federal de Minas Gerais, Belo Horizonte 31270-901, Brazil; giunchetti@icb.ufmg.br; 8Instituto Nacional de Ciência e Tecnologia de Doenças Tropicais (INCT-DT), Salvador 40110-160, Brazil; 9Department of Natural Sciences, University of South Carolina Beaufort, 1 University Boulevard, Bluffton, SC 29909, USA; edantonio@uscb.edu

**Keywords:** leishmaniasis, antileishmanial activity, malvidin, echioidinin, cytotoxicity, arginase inhibition

## Abstract

Leishmaniasis, a neglected tropical disease caused by *Leishmania* species, presents serious public health challenges due to limited treatment options, toxicity, high costs, and drug resistance. In this study, the in vitro potential of malvidin and echioidinin is examined as antileishmanial agents against *L. amazonensis*, *L. braziliensis*, and *L. infantum*, comparing their effects to amphotericin B (AmpB), a standard drug. Malvidin demonstrated greater potency than echioidinin across all parasite stages and species. Against *L. amazonensis*, malvidin’s IC_50_ values were 197.71 ± 17.20 µM (stationary amastigotes) and 258.07 ± 17 µM (axenic amastigotes), compared to echioidinin’s 272.99 ± 29.90 μM and 335.96 ± 19.35 μM. AmpB was more potent, with IC_50_ values of 0.06 ± 0.01 µM and 0.10 ± 0.03 µM. Malvidin exhibited lower cytotoxicity (CC_50_: 2920.31 ± 80.29 µM) than AmpB (1.06 ± 0.12 µM) and a favorable selectivity index. It reduced infection rates by 35.75% in *L. amazonensis*-infected macrophages. The in silico analysis revealed strong binding between malvidin and *Leishmania* arginase, with the residues HIS139 and PRO258 playing key roles. Gene expression analysis indicated malvidin’s modulation of oxidative stress and DNA repair pathways, involving genes like GLO1 and APEX1. These findings suggest malvidin’s potential as a safe, natural antileishmanial compound, warranting further in vivo studies to confirm its therapeutic efficacy and pharmacokinetics in animal models.

## 1. Introduction

Leishmaniasis, a vector-borne disease caused by the parasitic protozoa of the *Leishmania* genus, affects millions of people globally, particularly in tropical and subtropical regions. Spread by sandflies, it manifests in three main clinical forms [1]. The most severe, visceral leishmaniasis (VL), is characterized by fever, weight loss, and organ enlargement. The second clinical category, tegumentary leishmaniasis (TL), encompasses both cutaneous (CL) and mucosal leishmaniasis (ML) presentations. ML, the most destructive form, causes significant injuries [2]. Several *Leishmania* species are implicated in the disease. *L. infantum*, *L. amazonensis*, and *L. braziliensis* are some of the most prevalent in South America [3]. *L. infantum* has a broad geographic distribution, causing VL in regions like the Mediterranean, East Africa, Asia, and Latin America. This parasite primarily infects dogs in regions where the disease is a zoonosis, and can infect humans where it is an anthroponosis [4]. *L. amazonensis* demonstrates a wider range of clinical manifestations being responsible for both CL and ML [5] and it is also suggested as the cause of VL [6]. Finally, *L. braziliensis* is associated with a particularly aggressive form of CL. The lesions induced by *L. braziliensis* can be large, chronic, and lead to substantial scarring [7]. According to the World Health Organization, an estimated 0.7 to 1 million new leishmaniasis cases occur globally annually [8]. This neglected disease significantly impacts public health, causing disability, disfigurement, and even death if left untreated. Notably, 85% of global VL cases are reported from just the following seven countries: Brazil, Ethiopia, India, Kenya, Somalia, South Sudan, and Sudan [6,9]. The transmission process begins with a sandfly bite. Sandflies deposit *Leishmania* parasites in their promastigote form (extracellular, with a flagellum) into the mammalian host’s skin. Macrophages then phagocytose these parasites, the body’s first line of defense [10]. Inside a macrophage compartment for degrading engulfed particles (phagolysosome), the parasites transform into amastigotes (non-flagellated, replicative) [11]. *Leishmania* parasites employ a complex interplay of strategies to establish infection. They can suppress macrophage activation, hindering the production of pro-inflammatory cytokines (signaling molecules) that typically recruit and activate other immune cells to eliminate the parasite. This creates a permissive environment for parasite survival [12]. *Leishmania*’s repertoire of immune evasion strategies extends to molecular mimicry. Certain parasite species express molecules that closely resemble host components. This ingenious disguise allows for evading recognition and elimination by the immune system [13].

The phagolysosome presents a unique challenge for *Leishmania* due to its acidic environment, typically ranging from a pH of 4.5 to 5.5. However, amastigotes, the replicative form within the mammalian host, demonstrate a higher tolerance for this acidic niche. Amastigotes possess various mechanisms to adapt and thrive in harsh conditions [14]. Some studies have proposed that *Leishmania* might be able to manipulate the phagolysosome’s pH to create a more favorable environment for its growth [15,16,17]. This manipulation could involve the parasite actively expelling protons from the phagolysosome, reducing its acidity [18]. The precise mechanisms underlying *Leishmania*’s tolerance of the acidic environment and its potential manipulation of phagolysosomal pH remain under investigation. However, elucidating this intricate interplay between parasites and the environment holds immense potential for developing novel treatment strategies. By targeting the parasite’s ability to survive in the acidic environment or disrupting its pH manipulation mechanisms, researchers aim to disrupt its lifecycle and ultimately prevent successful infection.

Leishmaniasis treatment presents a complex challenge despite some recent improvements. While effective, the standard medications, such as pentavalent antimonials and amphotericin B (AmpB), have limitations that restrict their optimal use. These medications are associated with significant side effects [19]. Pentavalent antimonials cause severe reactions like nausea, vomiting, muscle pain, and irregular heartbeats. AmpB poses a challenging problem due to its nephrotoxicity and requires hospitalization for administration because of its toxicity [20]. These side effects can significantly impact a patient’s ability to adhere to the treatment regimen and complete the entire course, potentially leading to treatment failure. The efficacy of leishmaniasis treatment is additionally hindered by the variability in medication response among different *Leishmania* species and the coexistence of overlapping species [21]. Certain species have a differing susceptibility to the same drug, which poses a challenge in selecting the appropriate treatment. The emergence of drug-resistant *Leishmania* strains exacerbates this problem, presenting a substantial challenge to existing treatments [22]. These problems highlight the importance of developing new treatment approaches focusing on specific pathways and targets crucial for the parasite’s survival and growth inside the host. It is essential to consider the variation in species and the development of drug resistance when working to develop improved and safer treatments for leishmaniasis.

One promising approach involves targeting specific enzymes crucial for the parasite’s metabolism. *Leishmania* parasites rely on the enzyme arginase (ARG) for polyamine biosynthesis, a vital process for their growth and survival [23]. ARG has recently obtained considerable attention since new studies have highlighted it as a potential therapeutic target in leishmaniasis [24,25,26]. ARG is the first enzyme of the polyamine pathway. It catalyzes the conversion of L-arginine to L-ornithine and urea, and L-ornithine continues as the next metabolite in polyamine biosynthesis to form putrescine and spermidine [27]. Studies have shown that inhibiting ARG activity with specific compounds can effectively lead *Leishmania* parasites to cell death in laboratory studies (in vitro) and animal models. The inhibition results in a lack of protection against reactive oxygen species (ROS), which damages *Leishmania*’s genetic material. These examples highlight a few exciting new research directions in leishmaniasis treatment [28]. By targeting specific vulnerabilities in the parasite’s biology, researchers hope to develop safer, more effective therapies with minimal side effects and a reduced risk of resistance. The search for alternatives to traditional leishmaniasis medications has led researchers to explore natural products derived from plants and other organisms. These natural products often have a lower toxicity profile and might offer new avenues for combating the parasite. Our group’s recent research has been particularly encouraging, shedding light on the potential of specific natural compounds like malvidin (PubChem CID: 159287) and echioidinin (PubChem CID: 15559079) (Figure 1) [29].

Our previous study investigated malvidin using computational modeling predicting interactions with ARG which is vital for *Leishmania* survival and may disrupt parasite growth [29,30]. However, no in vitro or in vivo studies have yet confirmed its efficacy against *Leishmania*. Another promising natural product, echioidinin derived from *Andrographis echioides* and *Andrographis alata*, has demonstrated a cytotoxic effect on leukemic cell lines while showing low toxicity to mammalian cells, suggesting it could be a therapeutic alternative against leishmaniasis [30,31,32]. Echioidinin also exhibited strong antifungal activity against several fungi, potentially disrupting DNA synthesis and mitochondrial function, though the exact mechanisms remain under investigation [33,34]. A crucial aspect of exploring natural products, like malvidin and echioidinin, is ensuring their safety and computer-based methods play a vital role. These methods allow for rapid and cost-effective screening of various natural compounds [35]. Researchers can prioritize the most promising candidates for further evaluation by analyzing potential anti-leishmanial activity and predicted toxicity. In silico methods can also model the interactions between natural compounds and human biomolecules at an atomic level. This helps predict potential adverse effects by identifying how a compound might interact with essential proteins, enzymes, or DNA, potentially causing disruptions or damage [36].

Furthermore, in silico tools are invaluable for elucidating compounds’ potential mechanisms of action against *Leishmania* parasites. Researchers can design and optimize targeted therapies more effectively by understanding how natural products disrupt parasitic processes. In this study, the aim was to evaluate the in vitro inhibitory effects of malvidin and echioidinin on the ARG of three Leishmania species. Subsequently, in silico docking studies were conducted to analyze the binding interactions of the best candidate with the ARG at a pH of 5 and 7 by molecular dynamics (MD) simulations, assessing its behavior and stability. Also, network pharmacology and single-cell RNA sequencing data mining were used to analyze the potential molecular targets and pathways of NPs on different human organ datasets to predict potential tissue-specific toxicity. In this research, the promise of innovative enzyme inhibitors in expanding the field of drug discovery for leishmaniasis is highlighted and it provides critical molecular-level insights. The findings suggest that natural products could significantly advance the development of new therapeutic agents against leishmaniasis.

## 2. Results and Discussion

### 2.1. Antileishmanial Activity

In this study, the in vitro antileishmanial activity was evaluated of malvidin and echioidinin, two natural compounds, against three *Leishmania* species of *L. amazonensis*, *L. braziliensis*, and *L. infantum*. The experiments assessed the effectiveness against promastigote and axenic amastigotes forms [37]. AmpB, a well-established antileishmanial alternative, was used as positive control (Table 1).

Malvidin demonstrated consistently greater potency than echioidinin across both parasite stages and all three *Leishmania* species tested. The effectiveness of the compounds was measured using IC_50_ data, the concentration required to inhibit 50% of parasite growth, with lower IC_50_ values indicating higher potency. Malvidin exhibited lower IC_50_ values compared to echioidinin in all cases. The IC_50_ values for malvidin were as follows: 197.71 ± 17 µM for stationary promastigotes and 258.07 ± 17 µM for axenic amastigotes in *L. amazonensis*; 164.5 ± 19 µM for stationary promastigotes and 227.89 ± 17 µM for axenic amastigotes in *L. braziliensis*; and 141.26 ± 26 µM for stationary promastigotes and 173.86 ± 19 µM for axenic amastigotes in *L. infantum*. The IC_50_ values for *L. infantum* were slightly lower than the other species.

The IC_50_ values for echioidinin were as follows: 272.99 ± 29 µM for stationary promastigotes and 335.96 ± 19 µM for axenic amastigotes in *L. amazonensis*; 251.18 ± 22 µM for stationary promastigotes and 293.04 ± 20.76 µM for axenic amastigotes in *L. braziliensis;* and 212.83 ± 16 µM for stationary promastigotes and 261.73 ± 17µM for axenic amastigotes in *L. infantum*.

The IC_50_ values for AmpB were 0.08 ± 0.02 µM/0.09 ± 0.02 µM, 0.09 ± 0.03 µM/0.10 ± 0.03 µM, and 0.06 ± 0.01 µM/0.08 ± 0.02 µM against *L. amazonensis*, *L. braziliensis*, and *L. infantum* for stationary promastigotes/axenic amastigotes, respectively. Malvidin exhibited superior in vitro efficacy against stationary promastigotes compared to axenic amastigotes. Also, natural anthocyanins like malvidin are known for their diverse biological properties, including antiparasitic activity. Studies have shown that some flavonoids extracted from *Byrsonima coccolobifolia*, such as isoquercetin, catechin, and epicatechin, act as non-competitive inhibitors of ARG [38]. These flavonoids bind to sites other than the active site, disrupting the production of polyamines, which are crucial for parasite survival. Additionally, they may interact with parasite membranes, disrupt essential cellular processes, or impair *Leishmania*’s ability to survive within host cells [39,40].

### 2.2. Cytotoxicity Assay

The CC_50_ values were 2920.31 ± 80, 3010.62 ± 114, and 1.06 ± 0.1 µM for malvidin, echioidinin, and AmpB, respectively. The CC_50_ value, which indicates the concentration required to kill 50% of the cells, is an important measure of cytotoxicity. A higher CC_50_ value, as seen with malvidin, suggests lower toxicity to healthy cells, whereas a lower CC_50_ value, as seen with AmpB, indicates higher toxicity.

Using these data, the selectivity index (SI) was calculated by the ratio between the CC_50_ and IC_50_ values. For malvidin, the SI values were 14.70/12.60, 17.60/11.00, and 20.70/14.70 against *L. amazonensis*, *L. braziliensis*, and *L. infantum* for stationary promastigotes/axenic amastigotes, respectively. For echioidinin, the SI values were 10.80/8.90, 12.10/10.30, and 14.00/11.60 against *L. amazonensis*, *L. braziliensis*, and *L. infantum* for stationary promastigotes/axenic amastigotes, respectively. For AmpB, the SI values were 11.30/11.00, 12.70/9.80, and 16.70/12.60 against *L. amazonensis*, *L. braziliensis*, and *L. infantum* against the same species in the same stages. An SI value of 10 indicates high selectivity [41], implying that the compound is more toxic to the parasites than normal cells, making it a promising candidate for drug development. Consequently, these results show that malvidin has significant antileishmanial activity with lower mammalian cell toxicity than AmpB, as evidenced by its higher SI values. Among the *Leishmania* species tested, *L. infantum* showed the highest susceptibility to malvidin, reflected by the highest SI values.

These results are comparable to, and in some cases exceed, the selectivity indices reported by other authors for various natural compounds derived from different plant parts and essential oils. Compounds such as oleanic and maslinic acids [42], as well as those from *Connarus suberosus* [43], *Pseudelephantopus spiralis* [44], and certain species from Saudi Arabia [45], Mexico [46], and Brazil [47], have shown SI values ranging from 1 to 10. This suggests that malvidin demonstrates competitive or enhanced selectivity. None of the tested natural products significantly damaged the O^+^ human red blood cells (Table 1). The RBC_50_ values were determined to be 14.28 ± 1 µM for AmpB and 3120.13 ± 107 µM for malvidin. Consequently, malvidin exhibited significantly lower hemolytic activity compared to pure AmpB.

### 2.3. Treatment of Infected Macrophages

The efficacy of malvidin and echioidinin against *Leishmania* spp. was evaluated by measuring infection rates, the number of amastigotes per macrophage, and infectiveness reduction (Table 2). These assessments are crucial for determining how effectively these compounds inhibit the parasite’s lifecycle and may reveal underlying mechanisms of action. Additionally, this model allows comparative analysis with existing treatments and other experimental compounds. Strong efficacy results could justify further clinical investigations.

For *L. amazonensis*, malvidin exhibited a concentration-dependent reduction in infection rates. At the highest concentration (120.74 µM), the infection rate decreased to 59.30%, with the number of amastigotes per macrophage reduced to 3.80, resulting in a 35.75% reduction in infectiveness. Lower concentrations showed diminished efficacy. Echioidinin also demonstrated concentration-dependent effects, with the highest concentration (140.72 µM) resulting in a 63.40% infection rate.

For *L. braziliensis*, malvidin at 120.74 µM reduced infection rates to 39.80%, with 3.30 amastigotes per macrophage, resulting in a 42.07% reduction in infectiveness. Lower concentrations had less impact. Echioidinin at 140.72 µM led to a 43.20% infection rate, with 3.70 amastigotes per macrophage and a 37.12% reduction in infectiveness. Lower concentrations were less effective.

Regarding *L. infantum*, malvidin at 120.74 µM reduced infection rates to 49.80%, with 2.70 amastigotes per macrophage, achieving a 33.15% reduction in infectiveness. Lower concentrations were less effective. Echioidinin at 140.72 µM achieved a 50.70% infection rate and 4.20 amastigotes per macrophage, leading to a 31.95% reduction in infectiveness. AmpB was used as a control, showing a reduction in infectiveness of 73.78%, 75.98%, and 73.41% for *L. amazonensis*, *L. braziliensis*, and *L. infantum,* respectively.

Moreover, the treatment of infected macrophages was assessed to investigate the potential in vitro therapeutic efficacy of this compound against *Leishmania* spp. in mammalian cells. The results demonstrated that malvidin also exhibited a satisfactory selective index compared to the data obtained from using AmpB.

Screening of antileishmanial candidates is usually performed on parasite-stationary promastigote cultures, primarily due to the ease of cultivating the parasites and the resulting yield. However, evaluations using axenic amastigotes were also conducted since this parasite stage directly interacts with the host immune system and is responsible for the active disease [19]. Based on these in vitro results, malvidin emerged as the optimal candidate due to its superior SI and overall performance in key parameters, outperforming echioidinin in terms of selectivity and efficacy. Given this, we prioritized malvidin for further in silico analysis, aiming to explore its molecular interactions in more detail and better understand its potential as an antileishmanial agent.

### 2.4. In Silico Analysis

#### 2.4.1. Electrostatic Potential Surface and Frontier Orbitals of Malvidin and ABH Compounds

To fully understand the potential of malvidin (Figure 2A) for ARG inhibition, it is essential to explore its electronic properties and binding interactions comprehensively. In this study, the computational methods were leveraged to analyze malvidin’s electrostatic potential surface (EPS) and frontier molecular orbitals (FMOs). The EPS map will pinpoint regions on malvidin that could engage with the ARG active site. The FMO analysis, which focuses on the HOMO–LUMO molecular orbitals, will provide insight into its electron-donating and electron-accepting capabilities during binding.

In drug discovery, these computational approaches are important for characterizing the molecular interactions and binding affinities of potential drug candidates. The EPS map identifies regions of a molecule likely to participate in electrostatic interactions with target sites, such as ARG [48]. By highlighting areas with positive or negative charges, it identifies prospective sites for hydrogen bonding or ionic interactions with specific residues in the target’s active site. Simultaneously, analyzing the HOMO and LUMO of malvidin helps clarify its electronic behavior during binding, offering critical insights into its molecular interactions with the enzyme’s active site [49,50]. These analyses can guide structural modifications to enhance binding efficiency, stability, and selectivity.

We used ABH as a control molecule to assess the specificity of these interactions and gain insights into crucial functionalities for inhibition (Figure 2D). ABH has a well-defined structure with a boronic acid moiety known to participate in reversible covalent interactions with enzymes [51]. However, it lacks the extensive network of conjugated double bonds and aromatic rings present in malvidin. By comparing the electronic properties and docking results of malvidin and ABH, we can differentiate between general electrostatic interactions and specific interactions crucial for ARG inhibition by malvidin. This comparative approach will provide valuable insights for the targeted design of novel ARG inhibitors with improved potency and selectivity.

The ESP of malvidin (see Figure 2B) reveals a heterogeneous electron density distribution, with high-density regions (red) predominantly located on the lower methoxy group and some hydroxyl groups. The hydroxyl groups on the two conjoined rings show medium electron density (yellow/green), while low-density regions (blue) are near the hydrogen atoms. This distribution indicates that malvidin can form specific protein interactions through hydrogen bonds and electrostatic interactions. High electron density areas can interact with positively charged amino acid residues, while low-density regions can interact with negatively charged residues [52].

The HOMO of malvidin is distributed over aromatic rings and hydroxyl groups, suggesting that these sites are prone to donating electrons and acting as nucleophiles. The LUMO, also localized on the aromatic rings and functional groups, identifies electrophilic regions that can accept electrons (see Figure 2C). These properties suggest that malvidin can stabilize through specific interactions with amino acid residues in proteins, enhancing its affinity for active sites and its role in biological processes.

The ESP of ABH (see Figure 2E) shows a high electron density (red) on the two hydroxyl groups of the boronic acid moiety, in addition to the α-carboxylic acid group. The high electron density regions of the compound are either negatively charged or have permanent dipoles that produce a partial negative charge character (δ^−^). These electrostatic effects can enable ABH to engage in hydrogen bonding as well as electrostatic interactions with positively charged residue side chains. The medium electron density region (yellow/green) is spread along the aliphatic chain, indicating a more neutral charge distribution that can interact with hydrophobic regions of proteins. As expected, the low electron density (positive electrical field) (blue) is found in a smaller area of the molecule, such as the α-amino group. This observation suggests this region is more involved in participating in strong electrostatic interactions with negatively charged residue side chains of the enzyme’s active site. The HOMO of ABH is primarily localized around the amino group and adjacent carbon atoms, indicating that interacting residue side chains of the enzyme would act as nucleophilic sites capable of donating electrons to the amino group. On the other hand, the LUMO is mainly situated in the carboxyl group, indicating that interacting residue side chains would act as electrophilic sites and be capable of accepting electrons (see Figure 2F). This combination of nucleophilic and electrophilic characteristics allows ABH to interact with various amino acid residues in the active site of arginase, specifically a region well known as the L-α-amino acid recognition site and thus allowing for specific L-amino acid stereochemistry [53,54].

#### 2.4.2. Chemical Reactivity Properties of Malvidin and ABH Compounds

Malvidin and ABH exhibit distinct profiles in global chemical reactivity (Table 3). These key reactivity descriptors suggest distinct behaviors and potential biological impacts [55,56]. With its smaller energy gap and higher chemical potential, malvidin exhibits increased chemical reactivity and a pronounced tendency to attract electrons, facilitating robust interactions with the electron-rich and polar regions of proteins. The greater global softness of this compound indicates a high degree of electron cloud deformability [57], allowing for flexible binding to various protein sites, including those that are sterically hindered. The high electrophilicity index of malvidin suggests that it can readily accept electrons from nucleophilic amino acid residues, forming stable complexes that could significantly affect the activity and function of the protein. Additionally, the substantial dipole moment of malvidin enhances its interactions with polar regions of the protein, potentially leading to strong dipole–dipole interactions that further stabilize binding and influence protein conformation and dynamics [58].

Conversely, ABH, characterized by a larger energy gap and lower chemical potential for the mid-section of this chemical structure, demonstrates lower reactivity and a reduced tendency to transfer electrons, resulting in fewer and weaker interactions with proteins. ABH’s higher global hardness and lower global softness indicate a less flexible electronic structure, which limits its ability to adapt to various binding sites on the protein, thus reducing the likelihood of strong or specific interactions. The lower electrophilicity index of ABH suggests it is less prone to forming stable complexes with nucleophilic residues, diminishing its potential to alter protein function. Furthermore, the lower dipole moment of ABH indicates reduced polarity, which may limit its interactions with polar and charged regions of the protein, leading to minimal binding affinity and specificity.

Based on our computational analysis, malvidin appears to have high potential for engaging in intermolecular interactions in an enzyme’s active site based on its charge character, flexibility, and polarity. These interactions could have biological implications, such as modulating enzyme activity, altering signaling pathways, or affecting protein stability [59,60,61,62]. On the other hand, from a computational basis considering electrostatics alone, ABH appears to have higher stability and lower reactivity. These features may suggest that ABH is more modest in forming a protein–ligand complex, thus further suggesting a more inert behavior with minimal biological impact. However, this is indeed not the case with ABH and ARG. The boron chemistry involved with ABH allows for an interesting structure–analog inhibition case in the active site. The boron atom (of boronic acid) has a deficient octet by two electrons, and when ABH binds in the L-α-amino acid recognition site in ARG, the Y-shaped boronic acid moiety (showcasing *sp*^2^ hybridization) is immediately subject to nucleophilic attack by bridging hydroxide stemming off a Mn^2+^_2_ cluster. A tetrahedral boronate anion (*sp*^3^ hybridization) is produced that resembles the unstable *sp*^3^ hybridized transition state that L-arginine undergoes during catalysis. In the case of ABH, the boronate anion is highly stable and serves as one of the strongest arginase inhibitors known and may not be able to be predicted by a computational assessment.

If we consider a comparative analysis on primary electrostatics between ABH and malvidin, there appears to be more potential for malvidin to serve as a more effective modulator in biochemical applications, while ABH may be more suited for its more specific role requiring chemical transition state chemistry involving the boron atom. From a quantum analysis comparison of ABH and malvidin, malvidin reveals high reactivity and a pronounced tendency to attract electrons, facilitating robust interactions with proteins through hydrogen bonds and electrostatic interactions. Its high electron density areas allow it to interact with positively charged residue side chains, while its electronic flexibility allows greater adaptability to various protein binding sites. In contrast, ABH is characterized by a larger energy gap and lower chemical potential that demonstrates reduced reactivity and a less flexible electronic structure, limiting its specific interactions.

#### 2.4.3. Homology Modeling, Docking, and Molecular Dynamics of Leishmania ARG

The interaction between malvidin and ARG enzymes that catalyze the hydrolysis of arginine was studied in our previous work to understand the potential therapeutic effects of malvidin [29]. MD simulations analyzed the stability of the ARG–malvidin complex during an equilibrium simulation time of 100 ns. Table 4 shows the average root-mean squared deviation (RMSD) values, where an average value of less than 0.3 nm is observed, indicating that the enzyme–ligand complex is in equilibrium. The compactness of the ARGs was analyzed by calculating the radius of gyration (RG) and the solvent-accessible surface area (SASA). The average SASA value of the ARGs remained compact throughout the trajectory, meaning that their secondary and tertiary structures remained stable throughout the simulation without losing protein folding. This information can be observed especially through the values of the protein surface area, which did not show significant variations and remained constant around 333, 345, and 336 nm^2^ for *L. amazonensis*, *L. braziliensis*, and *L. infantum*, respectively. The RG values of the enzyme models did not vary in any of the simulations. This analysis allowed us to evaluate the structural dimension of the complexes where these measures remained constant throughout the trajectory in all simulations performed, confirming that the complexes remained compact and protein folding remained stable.

We employed the MM/GBSA method to determine the binding free energy from the trajectories obtained during the last 10 ns of MD simulations. The results, summarized in Table 5, highlight the interaction between ARG and malvidin at different pH levels. At a pH of 5, the total binding free energies (ΔG_TOTAL_) of malvidin were −9.73, −8.41, and −12.83 kcal/mol for *L. amazonensis*, *L. braziliensis*, and *L. infantum*, respectively. At a pH of 7, the ΔG_TOTAL_ values were −12.82, −12.05, and −10.85 kcal/mol for *L. amazonensis*, *L. braziliensis*, and *L. infantum*, respectively. These binding free energy values indicate a robust binding affinity of malvidin to ARG across the three *Leishmania* species, with slight variations in energy values between the different pH levels.

The consistently negative ΔG_TOTAL_ values across all systems and pH conditions suggest that the binding interactions are thermodynamically favorable. The release of energy upon binding indicates a stable complex formation between ARG and malvidin. The absence of significant fluctuations in the binding energies across different species and pH levels reinforces the stability and consistency of these interactions.

The data in Table 5 indicate that at a pH of 5, the most significant energetic contributions arise from the electrostatic contribution to free energy calculated by generalized born (ΔE_GB_) and polar solvation free energy (ΔG_SOLV_). Conversely, at a pH of 7, the primary energetic contributions are from electrostatic energy (ΔE_EL_) and gas-phase free energy (ΔG_GAS_). These findings suggest that the overall charge distribution of the systems varies with the pH, as illustrated in Figure 2. At a pH of 5, the ΔG_SOLV_ contributes more favorably to the binding free energy than at a pH of 7. In contrast, at a pH of 7, the ΔG_GAS_ contributes more to the binding free energy.

These results underline the pH-dependent nature of binding interactions, where different energetic components dominate under different conditions. At a pH of 5, the solvation effects (ΔG_SOLV_) are more prominent, suggesting that the solvation environment significantly influences binding. At a pH of 7, the dominance of gas-phase interactions (ΔG_GAS_) implies that the intrinsic electrostatic and van der Waals interactions play a more crucial role.

Understanding these pH-dependent energetic contributions is essential for optimizing ligand binding in drug design. It highlights the need to consider environmental factors such as pH when developing inhibitors targeting ARG in *Leishmania* species. Future experimental studies should aim to validate these computational insights and explore the implications for therapeutic efficacy under varying physiological conditions.

Our molecular docking, MD, and energetic contribution study investigated the potential of malvidin to inhibit ARG enzymes from the three *Leishmania* species of *L. amazonensis*, *L. braziliensis*, and *L. infantum*. This represents the first detailed analysis of malvidin’s interactions with the active sites of these enzymes at two relevant pH levels (a pH of 5 and a pH of 7). The docking simulations revealed key amino acid residues involved in malvidin’s binding, which varied slightly between species and pH levels but consistently interacted with critical functional groups. Additionally, electrostatic potential surface calculations confirmed that protonation states significantly influenced the overall charge of the ARG models. In Figure 3, at a pH of 5 (blue color), ARG exhibited a stronger positive electrostatic potential due to the protonation of basic residues, while at a pH of 7 (red color), a more pronounced negative charge was observed. Despite these differences, malvidin remained bound to the ARG enzyme at both pH levels, demonstrating stable interactions. In this comprehensive study, valuable insights are provided into malvidin’s binding mechanisms and its potential as a therapeutic agent against different *Leishmania* species.

For *L. amazonensis* (Appendix A), at a pH of 5, malvidin interacted with HIS 139, ASP 141, ASN 143 (carbon hydrogen bond), HIS 154, GLY 155 (carbon hydrogen bond), ALA 192, ASP 194, GLU 197, THR 257 (carbon hydrogen bond), PRO 258 (conventional hydrogen bond), VAL 259, ARG 260, and GLU 288. At a pH of 7, the interactions included HIS 139 (pi-pi stacked and pi alkyl), ASP 141 (carbon–hydrogen bond), ILE 142, ASN 143 (conventional and carbon–hydrogen bonds), SER 150, HIS 154 (pi alkyl and carbon–hydrogen bond), GLY 155 (carbon–hydrogen bond), ALA 192, VAL 193, ASP 194 (carbon–hydrogen bond), GLU 197, THR 257, PRO 258, VAL 259, and ARG 260. Notably, the residues HIS139, ASN143, SER150, GLY155, ALA192, GLU197, THR257, PRO258, and VAL259 are significant, with HIS139, GLY155, and THR257 displaying energy values above −1 kcal/mol.

In the case of *L. braziliensis* (Appendix A), at a pH of 5, the interacting amino acids were HIS 140 (conventional hydrogen bond and pi-pi stacked), ASP 142, ASN 144, HIS 155 (pi-alkyl), GLY 156, ASP 193, GLU 195, GLU 198, THR 258, PRO 259, VAL 260, and ARG 261 (unfavorable donor-donor). At a pH of 7, the interactions included HIS 140 (pi-pi stacked and conventional hydrogen bond), ALA 141, ASP 142 (carbon–hydrogen bond), ILE 143, ASN 144, SER 151, HIS 155 (pi alkyl), GLY 156 (carbon–hydrogen bond), ASP 193 (pi anion), GLU 195, GLU 198 (carbon and conventional hydrogen bonds), THR 248, PRO 259, VAL 260, ARG 261, and GLU 289. The residues HIS140, SER151, GLY156, GLU195, GLU198, THR258, PRO259, and VAL260, with HIS140 and THR258 showing energy values above −1 kcal/mol.

For *L. infantum* (Figure 4), at a pH of 5, the interacting amino acids were HIS 139 (conventional hydrogen bond and pi cation), ALA 140, ASP 141, ASN 143 (conventional hydrogen bond), SER 150, ASN 152, HIS 154 (pi-alkyl), GLY 155, ASP 194 (carbon–hydrogen bond), GLU 197 (conventional hydrogen bond), THR 257 (carbon–hydrogen bond), PRO 258 (conventional hydrogen bond), VAL 259, and ARG 260. At a pH of 7, the interactions included HIS 139 (pi cation), ASP 141 (carbon–hydrogen bond), ILE 142, ASN 143, SER 150, HIS 154 (pi alkyl), GLY 155 (carbon–hydrogen bond), ALA 192, VAL 193, ASP 194 (carbon–hydrogen bond), GLU 197 (conventional hydrogen bond), THR 257 (carbon–hydrogen bond), PRO 258 (conventional hydrogen bond), VAL 259, ARG 260 (conventional hydrogen bond), and GLU 288. The key residues contributing to the binding energy were HIS139, ASN143, SER150, GLY155, VAL193, ASP194, GLU197, THR257, PRO258, and VAL259, with HIS139 and PRO258 exhibiting the highest contributions (>−1 kcal/mol).

These findings highlight the specific amino acids that play significant roles in interacting with malvidin across different *Leishmania* species. The residues HIS139 and PRO258 were consistently significant in *L. amazonensis* and *L. infantum*, while HIS140 and THR258 were prominent in *L. braziliensis* (Appendix A).

These findings suggest that malvidin may inhibit *Leishmania* ARG activity by interacting with critical amino acid residues. Notably, HIS 139, a critical binding site for malvidin across all *Leishmania* species, is conserved in ARG family enzymes involved in the coordination of the binuclear manganese cluster (Mn^2+^-Mn^2+)^ in the active site [65,66]. Blocking this metal coordination with other molecules has demonstrated the inhibition of ARG activity. Therefore, malvidin’s interaction with HIS 139 is likely a pivotal mechanism for its inhibitory effect [67]. This is supported by other studies, such as involving fisetin, which also showed enhanced stability through similar interactions [68]. Another significant interaction involves HIS 154, which directly binds the L-arginine substrate in some *Leishmania* ARGs. Malvidin’s interaction with HIS 154 could potentially hinder substrate binding, contributing to ARG inhibition [69]. Additionally, malvidin interacts with ALA 192, a residue previously targeted by other ARG inhibitors like phenylacetamide, ABH, and caryophyllene oxide. Since ALA 192 is not conserved in human ARG, this suggests a potential shared inhibitory mechanism between malvidin and these compounds [70,71,72]. Furthermore, some flavonoids, such as epicatechin, catechin, and gallic acid, have also been tested against ARG. Their interactions with additional residues, such as ASP 141, GLY 155, ASP 181, ASP 194, and THR 257, were observed [73]. Similar compounds, including polyphenols such as caffeic and rosmarinic acids, also interacted with HIS 139, ASP 194, and SER 150 [67].

These in silico findings provide a promising foundation for further research. In vitro and in vivo studies are necessary to confirm malvidin’s ability to inhibit *Leishmania* ARG and evaluate its efficacy against the parasite. Additionally, exploring the structure–activity relationship of malvidin and its derivatives could lead to developing more potent ARG inhibitors for treating Leishmaniasis.

##### Network Analysis of Protein–Protein Interactions and Single-Cell RNAseq Analysis in Key Organ Systems for Potential Toxicity

The integration of the network analysis of protein–protein interactions with single-cell RNA sequencing (scRNAseq) is transforming our understanding of drug-induced toxicity across key organ systems. Network analysis offers a system-level view of complex protein interactions within cellular pathways [74]. This approach enables the identification of critical nodes and pathways that may serve as potential drug targets or toxicity markers, particularly those implicated in processes like inflammation, oxidative stress, and cellular metabolism, which are often linked to adverse effects. Meanwhile, scRNAseq analysis offers a high-resolution view of gene expression at the cellular level, uncovering cellular heterogeneity and specific responses to toxic agents that are often obscured in bulk tissue studies [75]. By integrating these methods, researchers can link alterations in protein interactions with gene expression changes at the cellular level, offering a comprehensive understanding of how toxicity develops across different organ systems. This combined approach improves the prediction of early toxicity prediction—a major challenge in pharmaceutical research. This method highlights both established and novel drug-pathway connections, leveraging big data to improve drug safety and guide the development of therapies with a reduced risk of adverse effects in clinical trials [76,77]. This integrative approach is illustrated in the network visualizations shown in Figure 5A which depict the intricate interplay between proteins within the specific tissues of the brain, heart, kidney, and liver. Each graph represents a tissue-specific protein interaction network, emphasizing key tissue-specific toxicity markers (represented as circles) and predicted targets (represented as squares) based on their centrality within the network. The color intensity scale (yellow to dark blue) indicates the relative importance of each node in terms of its role and action within the network.

In the brain network, critical markers are involved in neuronal function and implicated in neurodegenerative diseases. These include acetylcholinesterase (ACHE), apolipoprotein E (APOE), amyloid precursor protein (APP), brain-derived neurotrophic factor (BDNF), cyclin-dependent kinase 5 (CDK5), Huntingtin (HTT), nerve growth factor (NGF), Parkinson protein 7 (PARK7), prion protein (PRNP), and presenilin 1 (PSEN1). The intricate interactions among these markers influence cell survival and contribute to the complex pathology of neurodegenerative diseases such as Parkinson’s, Alzheimer’s, Huntington’s, and Creutzfeldt-Jakob disease [78]. The heart network, on the other hand, features essential markers for maintaining cardiac function. Markers such as cysteine and glycine-rich protein 3 (CSRP3), myosin-binding protein C (MYBPC), telethonin (TCAP), tropomyosin 1 (TPM1), and titin (TTN) play crucial roles in maintaining cellular homeostasis within the heart muscle. Dysregulation of these genes can lead to various cardiac disorders, emphasizing their importance in negatively regulating cell death pathways [79].

The liver network emphasizes markers responsible for critical cellular defense mechanisms, neurotransmitter metabolism, and steroid hormone processing. Genes such as ATP-binding cassette subfamily C member 2 (ABCC2), ATP-binding cassette subfamily G member 2 (ABCG2), cytochrome P450 1A1 (CYP1A1), epoxide hydrolase 1 (EPHX1), UDP glucuronosyltransferase 1 family, polypeptide A10 (UGT1A10), UDP glucuronosyltransferase 1 family, polypeptide A3 (UGT1A3), and UDP glucuronosyltransferase 1 family, polypeptide A7 (UGT1A7) collectively contribute to liver detoxification, homeostasis, and metabolic regulation [80]. The kidney network highlights markers crucial for kidney development and function. Markers such as ataxia telangiectasia mutated (ATM), GATA binding protein 3 (GATA3), renin (REN), Ret proto-oncogene (RET), and tuberous sclerosis complex 2 (TSC2) play essential roles. Dysregulation of these genes can lead to structural abnormalities, impaired renal function, and electrolyte imbalances. Understanding their roles is vital for managing kidney health and developing targeted therapies [81,82].

In this context, several predicted malvidin targets were identified as central nodes within the protein interaction networks across key organ systems. These central nodes, due to their critical roles in maintaining the structural and functional integrity of key pathways, did not exhibit strong associations with potential toxic effects. Their centrality likely contributes to a stabilizing effect, where malvidin-induced perturbations are absorbed or buffered by the system, minimizing disruptions that could lead to toxicity. This observation highlights the significance of network topology when evaluating drug safety, as central proteins may play essential roles in cellular homeostasis without directly contributing to toxicity. These findings demonstrate the significance of integrating network analysis and scRNAseq to offer a comprehensive, system-level comprehension of drug-induced toxicity, hence guiding the development of safer therapies with reduced risks of side effects.

#### 2.4.4. Pathway Analysis of Key Genes in Different Tissues

Figure 5B illustrates the functional pathways associated with key proteins in each tissue type. The brain’s pathways involve proteins like monoamine oxidase A (MAOA), cyclin-dependent kinase 1 (CDK1), and xanthine dehydrogenase (XDH). These proteins play crucial roles in regulating cellular processes, impacting cell survival, development, and overall health. MAOA influences neurotransmitter levels [83], CDK1 orchestrates critical cell cycle events [84], and XDH generates reactive oxygen species [85]. The dysregulation of these proteins can lead to structural abnormalities, impaired function, and pathogenesis. Heart-specific pathways include interactions between thrombin (F2), AXL receptor tyrosine kinase, CDK1, endothelin-1 (EDN1), and XDH. These proteins play crucial roles in regulating the cellular processes within the heart. F2 plays a dual role in blood clotting and anticoagulation [86]. AXL influences cell survival and resistance to chemotherapy [87]. CDK1 orchestrates critical cell cycle events, maintaining genome stability [88]. EDN1 impacts vasoconstriction, inflammation, and tissue remodeling [89]. XDH generates oxidative stress and cellular health [90]. Kidney pathways feature proteins such as AXL, DNA topoisomerase 2 Alpha (TOP2A), BRCA1-associated protein 1 (BAP1), XDH, and F2. These proteins play crucial roles in regulating cellular processes within the kidney. AXL influences cell survival and tissue repair [91], TOP2A maintains genome stability [92], BAP1 is associated with cancer predisposition [93], XDH impacts purine metabolism and oxidative stress [94], and F2 plays a dual role in blood clotting and anticoagulation [95]. Liver pathways involve proteins like XDHs [96], aldo-keto reductase family 1 member C1 (AKR1C1) [97], and arachidonate 5-lipoxygenase (ALOX5) [98]. XL influences cell survival and tissue repair, TOP2A maintains genome stability [99], BAP1 is associated with cancer predisposition [100], XDH impacts purine metabolism and oxidative stress, and F2 plays a dual role in blood clotting and anticoagulation, highlighting the liver’s role in detoxification and metabolic regulation [101].

The findings suggest that although the identified marker targets for malvidin are central within their respective networks, they are not involved in processes critical for cellular function and homeostasis that could lead to toxicity if interrupted. Their centrality indicates that while these targets play important roles in their networks, their interaction with malvidin is unlikely to disrupt critical biological pathways or cause adverse effects and toxicity.

#### 2.4.5. Heatmap Analysis of Key Differentially Expressed Genes

The uniform manifold approximation and projection (UMAP) plot (Figure 5C) demonstrates distinct clustering of brain (red), heart (green), kidney (blue), and liver (purple) tissues based on their gene expression profiles. Noticeably, shifts in the clustering patterns of liver and kidney tissues suggest significant tissue-specific effects.

The heatmap (Figure 5D) displays the average expression levels of selected genes across different tissues following malvidin treatment. Interestingly, no significant overexpression of genes was observed in the brain. In the heart, malvidin treatment led to increased expression of AXL, a receptor tyrosine kinase known to be involved in cell survival, proliferation, migration, and angiogenesis [91]. However, AXL is often overexpressed in cancer cells and associated with poor prognosis [102]. Further investigation is needed to understand the specific role of AXL upregulation in the heart following malvidin treatment.

The kidney exhibited an increased expression of MAOA following malvidin treatment. Although primarily found in the brain, MAOA is also expressed at low levels in other tissues, including the kidneys. Studies have shown an association between lower levels of MAOA expression in the kidneys and chronic kidney disease (CKD) [103]. Increased MAOA expression in response to malvidin may offer potential benefits for kidney health, but further research is warranted.

In the liver, malvidin treatment resulted in the altered expressions of several genes, including MAOA, F2, and AKR1C1. F2, which can be interpreted in two ways within the context of the liver, plays a dual role in blood clotting. The liver produces F2 and its expression is monitored to assess the post-transplantation functionality of donated livers [104]. However, a higher level of F2 can also indicate stage 2 fibrosis, a condition characterized by scar tissue development in various liver diseases [105]. In this case, F2 likely reflects the state of the liver tissue rather than direct gene expression changes. AKR1C1 encodes an enzyme involved in the body’s metabolic processes [106]. The research suggests that AKR1C1 might be overexpressed in liver cancer compared to healthy tissue, potentially serving as a disease marker or even contributing to its development [107].

Two genes, GLO1 and APEX1, were notably expressed across the heart, kidney, and liver. GLO1 acts as a detoxifying enzyme, specifically targeting a harmful molecule called methylglyoxal produced during normal cellular processes [108]. By removing methylglyoxal, GLO1 protects cells from damage and contributes to overall cellular health. APEX1, on the other hand, plays a critical role in DNA repair by removing damaged building blocks from DNA, allowing proper repair mechanisms to function [109]. Damaged DNA can lead to mutations and cell death, highlighting the importance of APEX1 in maintaining cellular health [110]. Similar to GLO1, APEX1 is expressed in various tissues and offers specific benefits to each organ. In the heart, APEX1 protects heart muscle cells from DNA damage linked to heart disease [111]. Similarly, it helps maintain kidney DNA integrity and potentially prevents kidney disease. Finally, in the liver, APEX1 is involved in DNA repair processes essential for detoxification and regeneration functions [111].

The combined analysis of the UMAP plot and heatmap provides valuable insights into the tissue-specific effects of malvidin treatment. The observed changes in gene expression patterns suggest potential benefits for various organs, including the reduction in oxidative stress in the liver and the upregulation of DNA repair mechanisms in the heart, kidney, and liver. However, further research is needed to fully understand the underlying mechanisms and potential therapeutic applications of malvidin.

It is important to acknowledge that UMAP, while a powerful tool for visualizing high-dimensional data, introduces some degree of distortion into the underlying data structure. Therefore, further bioinformatic analysis methods, such as principal component analysis (PCA) or t-distributed stochastic neighbor embedding (t-SNE), could better understand gene expression patterns across these tissues. These UMAP visualizations provide a valuable starting point for dissecting tissue-specific gene expression patterns. These findings can guide future studies utilizing techniques like RNA-seq and quantitative PCR (qPCR) to validate candidate genes and explore their functional roles in distinct organ systems [112]. Understanding these complex transcriptional landscapes is crucial for advancing our knowledge of organ development, physiology, and disease pathogenesis.

In the bottom row of Figure 5E, UMAP visualizations are presented for gene expression data from the brain (red), heart (green), kidney (blue), and liver (purple) tissues. UMAP, a dimensionality reduction technique, projects high-dimensional gene expression data into a two-dimensional space, allowing for the visualization of differentially expressed genes across these tissues. The UMAP plots demonstrate clear segregation between tissue types, suggesting distinct transcriptional landscapes across organs. This separation likely reflects the expression of genes critical for each tissue’s specialized functions.

The heart UMAP might group genes encoding proteins involved in sarcomere function, electrical conduction, and cardiac rhythm regulation. Examples include genes for sarcomere function like myosin heavy chain 6 (MYH6) [113], alpha skeletal muscle actin (ACTA1) [114], troponin I type 2 (TNNI2) [115]; electrical conduction genes like sodium voltage-gated channel alpha subunit 5 (SCN5A) [116], potassium voltage-gated channel subunit (KCNQ1) [117], gap junction protein, alpha 1 (GJA1) [118]; and cardiac rhythm regulation genes like potassium channel, inwardly rectifying subfamily J member 11 (KCNJ11) [119], calcium channel, voltage-dependent, L-type, and alpha 1C subunit (CACNA1C) [120].

Kidney tissue may show enrichment for genes related to solute transport, electrolyte homeostasis, and waste product excretion. Examples include genes for solute transport like solute carrier family 2 (facilitated transporter) [121], aquaporin 1 (AQP1) [122], sodium–potassium–chloride cotransporter 1 (SLC12A1) [123]; electrolyte homeostasis genes like sodium–potassium-ATPase subunit alpha 1 (ATP1A1) [124], aquaporin 2 (AQP2) [125], chloride channel accessory 1 (CLCA1) [126]; and waste product excretion genes like solute carrier family 23 member 1 (SLC23A1) [127], multidrug resistance protein 1 (MDR1) [128], and uromodulin (UMOD) [129].

The liver UMAP may cluster genes implicated in xenobiotic detoxification, metabolic pathways, and bile acid synthesis. Examples include genes for xenobiotic detoxification like cytochrome P450 family 3 subfamily a member 4 (CYP3A4) [130], glutathione S-Transferase alpha 1 (GSTA1) [131], UDP glucuronosyltransferase 1 family member A1 (UGT1A1) [132]; metabolic pathway genes like glucokinase (GCK), fatty acid synthase (FASN), carnitine palmitoyltransferase 1A (CPT1A) [133]; and bile acid synthesis genes like cholesterol 7A-Hydroxylase (CYP7A1) [134], bile acid coenzyme A: amino acid N-Acetyltransferase (BAAT) [135], and sterol 12α-hydroxylase (CYP8B1) [136].

Further in-depth analysis of differentially expressed genes within each UMAP plot is warranted to identify tissue-specific markers and elucidate the underlying molecular mechanisms governing organ-specific functionalities. Functional enrichment analysis and gene ontology term association can be employed to identify biological processes and pathways significantly associated with each tissue cluster [137,138]. Additionally, co-expression network analysis can reveal potential interactions and regulatory relationships between differentially expressed genes within each tissue type. These UMAP visualizations provide a valuable starting point for dissecting tissue-specific gene expression patterns. These findings can guide future studies to unravel the complex interplay between genes and their functional roles in distinct organ systems. Understanding these complex transcriptional landscapes is crucial for advancing our knowledge of organ development, physiology, and disease pathogenesis.

## 3. Materials and Methods

### 3.1. Parasites

*L. amazonensis* (IFLA/BR/1967/PH-8), *L. braziliensis* (MHOM/BR/1975/M2903), and *L. infantum* (MHOM/BR/1970/BH46) were used as parasites. Stationary promastigotes were grown at a 24 °C in complete Schneider’s medium (Sigma-Aldrich, St. Louis, MO, USA), which was composed of medium plus 20% (*v*/*v*) heat-inactivated fetal bovine serum (FBS; Sigma-Aldrich, USA) and 20 mM L-glutamine with a pH of 7.4 [139]. To obtain the axenic amastigotes, 10^9^ stationary promastigotes were washed three times in sterile phosphate-buffered saline (PBS 1X and incubated in 5 mL FBS for 48 h (*L. amazonensis* and *L. braziliensis*) and 72 h (*L. infantum*), respectively, at 37 °C. Parasites were washed in cold PBS 1x, and their morphology was evaluated after staining using the Giemsa method in an optical microscope, as previously described [140].

### 3.2. Antileishmanial Activity

*L. infantum* IC_50_ was evaluated by incubating parasite stationary promastigotes or axenic amastigotes (10^6^ cells, each) with malvidin (MolPort catalogue N° AA00E9KS, Latvia) (0 to 150.92 µM) or echioidinin (MolPort catalogue N°AK-693/21087015, Latvia) (0 to 178.3 µM) in 96-well culture plates (Nunc, Nunclon, Roskilde, Denmark) for 48 h at 24 °C. AmpB (0 to 10.8 µM; Sigma-Aldrich, USA) was used as control. Cell viability was assessed through 3-(4.5-dimethylthiazol-2-yl)-2.5-diphenyl tetrazolium bromide (Sigma-Aldrich, USA) method. Optical density (OD) values were measured in a microplate spectrophotometer (Molecular Devices, Spectra Max Plus, San Jose, CA, USA) at 570 nm. IC_50_ values were calculated by sigmoidal regression of dose–response curves in Microsoft Excel software (version 10.0) [141]. Two-tailed Student’s *t*-tests (*p* < 0.05) were used to compare IC50 values between malvidin and echioidinin, assessing significant differences in inhibitory activity.

### 3.3. Cytotoxicity Assay

Cytotoxicity was evaluated ex vivo in murine macrophages and human red blood cells, for which concentrations inhibiting 50% of macrophages (CC_50_) and red blood cells (RBC_50_) were determined. Briefly, murine cells (5 × 10^5^) or a 5% human red blood cells suspension were incubated in the presence of malvidin (0 to 301.84 µM) or echioidinin (0 to 356.76 µM) AmpB (0 to 10.82 µM) was used as control. Cells were incubated in RPMI 1640 medium for 48 h (murine macrophages) or 1 h (red blood cells) at 37 °C in 5% CO_2_. Macrophage viability was assessed using the MTT method. The red blood cell suspension was centrifuged at 1000× *g* for 10 min at 4 °C, after which the percentage of cell lysis was evaluated spectrophotometrically at 570 nm. The hemolysis was determined by replacing the natural products with an equal volume of phosphate-buffered saline with a pH of 7.4 (PBS—negative control) or distilled water (positive control) [141]. CC_50_ and RBC_50_ values were calculated by sigmoidal regression of dose–response curves in Microsoft Excel software (version 10.0). SI was calculated as the ratio between CC_50_ and IC_50_ values.

### 3.4. Treatment of Infected Macrophages

To evaluate the efficacy of malvidin or echioidinin for treating infected macrophages, murine cells (5 × 10^5^) were plated on round glass coverslips in 24-well plates in RPMI 1640 medium supplemented with 20% (*v*/*v*) FBS and 20 mM L-glutamine at a pH of 7.4 and incubated for 24 h at 37 °C in 5% CO_2_. Stationary promastigotes were added to the wells at a ratio of 10 parasites per macrophage, and cultures were incubated further for 48 h at 37 °C for 5% CO_2_. Free parasites were removed by extensive washing with RPMI 1640 medium, and infected macrophages were treated with malvidin (30.18, 60.37, and 120.74 µM), echioidinin (35.18, 70.36, and 140.72 µM), or AmpB (2.70 and 5.41 µM) for 48 h at 24 °C in 5% CO_2_. After fixation with 4% paraformaldehyde, cells were washed and stained with Giemsa, and the infection percentage, the number of amastigotes per infected macrophage, and the reduction in the infection percentage were determined by counting 200 cells in triplicate, using an optical microscope [141].

### 3.5. In Silico Analysis

#### 3.5.1. Molecular Geometry Optimization

Natural products and 2(S)-amino-6-boronohexanoic acid (ABH, CID 9793992) SDF files were selected from the PubChem database [142]. ABH was used as a control because it is known to have an inhibitory effect on arginase [36], allowing for a comparison of its effects. The molecular geometries of these compounds were optimized using the B3LYP/6-311++g(d,p) basis set, incorporating implicit water solvent effects (CPCM(WATER)) [143,144], as implemented in ORCA software version 5.0 [145,146]. Vibrational analyses were performed to ensure accurate ground-state geometries and confirm the absence of imaginary frequencies. Atomic charges were calculated using the Hirshfeld population analysis to account for the electrostatic effects of the ligands within the protein complexes [147]. These charges were then used to reparametrize the AMBER-ILDF force field, which was subsequently used for molecular dynamics simulations.

#### 3.5.2. Global Reactivity Descriptors

Global reactivity descriptors were calculated using conceptual density functional theory (DFT) [148,149], which explains the relationships among structure, stability, and reactivity [150]. These descriptors were derived using Koopmans’ theorem which evaluates the energies of the highest occupied molecular orbital (HOMO) and lowest unoccupied molecular orbital (LUMO) obtained from simple-point energy calculations using B3LYP/6-311++g(d,p) basis set [151,152] (see Table 6). This analysis provided insights into how the molecules might interact with other molecules.

#### 3.5.3. Homology Modeling, Docking, and Molecular Dynamics of Leishmania ARG

Homology modeling was performed to generate 3D models of ARG from protein sequences (https://www.uniprot.org/, accessed on 15 May 2024) of *L. amazonensis* (AAC95287.1), *L. braziliensis* (KAI5689219.1), and *Leishmania infantum* (CAC9543944.1), using the Swiss-Model server (https://swissmodel.expasy.org, accessed on 10 August 2023) with ARG from *L. mexicana* (PDB code: 4ITY) as the template [72]. Seven high-resolution X-ray crystal structures of ARG from *L. mexicana* (1.95 Å or better) were also referenced, including the inhibitorless form (PDB 4ITY), complexes with competitive inhibitors (PDB 4IU0, 4IU1, 4IU4, 5HJ9, 5HJA), and with the reaction product L-ornithine (PDB 4IU5) [54,71]. These structures provided insights into the active site, Mn^2+^_2_-binuclear cluster, metal ion coordination, and zones of surface electrostatic charge. Structural differences were also compared against human arginase I to gain further insight into developing Leishmania ARG-selective inhibitors [54]. Sequence alignments using BLAST (https://blast.ncbi.nlm.nih.gov/, accessed on 27 December 2024) revealed that ARG from *L. amazonensis*, *L. infantum*, and *L. braziliensis* differed by 2, 13, and 40 residues, respectively, compared to *L. mexicana* ARG (GenBank accession no. AAR06176.1). The Amber topology file for malvidin was generated with the Acpype server (https://www.bio2byte.be/acpype/, accessed on 21 August 2023) with a total charge of +1. High-resolution crystallographic data and sequence comparisons were used to strengthen computational docking predictions, particularly for species with high sequence similarity [156,157].

Molecular docking was performed using the DockThor server (https://www.dockthor.lncc.br/v2/, accessed on 7 September 2023), where grids were set in the active site of the chain A of each protein. Complexes with the best scores were selected for further analysis. Protonation states of ionizable residues were determined using the PROPKA v. 3 program [158,159]. The summary of pKa values was used to calculate the protonation states of Asp, Glu, Arg, Lys, and His residues, as well as the C-terminal and N-terminal ends. The semi-grand canonical Monte Carlo (SGCMC) procedure was used to calculate the different protonation states at a pH of 5 and a pH of 7, based on the free energy associated with the pKa of each titratable residue. This was calculated using the program developed by Barazorda et al., 2021, accessible online at https://github.com/smadurga/Protein-Protonation accessed on 10 September 2023 [160,161].

After the protein preparation, MD simulations were used to research the ARG-malvidin complexes. Simulations were conducted using the AMBER-ILDF force field in GROMACS v. 2023.4 software [162]. The systems were then solvated with the TIP3P water model, and sodium (Na^+1^) or chloride (Cl^−1^) ions were added to neutralize the system. The simulation box size was set to 12 Å × 12 Å × 12 Å. Before running the MD simulations, an energy minimization step was performed using the steep-descent algorithm for 20,000 steps. The MD simulation itself was conducted in the following two distinct stages: the first stage was in the canonical NVT ensemble considering position restraint of the backbone and the coordination of the metal in the active site with distance restraint, with a trajectory of 10 ns. The second stage was the production dynamics in the isothermal–isobaric NPT ensemble with a simulation time of 100 ns. The V-rescale thermostat maintained the temperature at a physiological value of 300 K, and the Parrinello–Rahman barostat maintained pressure at 1 bar, reflecting typical cellular pressure conditions.

Finally, the binding free energy estimation between ARG and malvidin was calculated using the molecular mechanics generalized born surface area (MM/GBSA) method. This calculation employed data from GROMACS simulation and was performed using a tool based on AMBER’s MMPBSA.py script (gmx_MMPBSA v1.6.2 based on MMPBSA version 16.0 and AmberTools 20).

#### 3.5.4. Prediction of Natural Product Targets and Retrieval of Tissue Toxicity Markers

The natural product’s simplified molecular-input line-entry system (SMILES) codes were searched and retrieved from the PubChem server (https://pubchem.ncbi.nlm.nih.gov/, accessed on 21 March 2024) [163] and uploaded to the SwissTarget Prediction server (http://www.swisstargetprediction.ch/, accessed on 21 March 2024) [164], and to the Superpred server (https://prediction.charite.de/, accessed on 21 March 2024) [165], for target prediction on *Homo sapiens* species. To identify toxicity markers in the brain, heart, kidney, and liver tissues, the GeneCards database (https://www.genecards.org/, accessed on 21 March 2024) [166] was used, whereas the related protein/gene targets were retrieved using the keywords of “neurotoxicity”, “cardiotoxicity”, “nephrotoxicity”, and “hepatotoxicity”, respectively. The standard of proteins and gene names were obtained from the UniProt database (https://www.uniprot.org/, accessed on 15 April 2024) [167], and Venn diagrams were used to identify duplicate entries using the Draw Venn Diagram tool (http://bioinformatics.psb.ugent.be/webtools/Venn/, accessed on 15 April 2024).

#### 3.5.5. Protein–Protein Interaction Network Analysis

The predicted targets and tissue toxicity-associated markers were used to construct protein–protein interaction (PPI) networks. The molecular networks were retrieved from the STRING database (https://string-db.org, accessed on 20 April 2024) [168] and analyzed using the Cytoscape platform v 3.10.3 [169]. The cytoHubba v 0.1plugin was used to score and rank network nodes based on the maximal clique centrality (MCC) topological analysis method [170]. Cytoscape default settings were considered to visualize the network, whereas node type and color were manually adjusted, considering the scores provided by the MCC analysis and the type of marker. Finally, the plugin StringApp v 2.2.0 [171] was used to retrieve functional enrichment for gene ontology (GO) terms regarding biological processes (BPs), molecular functions (MFs), and cellular components (CCs), with results displayed in circular radial layouts.

#### 3.5.6. Single-Cell RNA Sequencing Analysis

The gene raw counts or normalized gene expression matrix of single-cell gene expression data of 04 relatively normal tissues were downloaded from the Gene Expression Omnibus (GEO) (https://www.ncbi.nlm.nih.gov/geo/, accessed on 30 October 2024). Kidney, GEO Accession No. GSE131685, from 3 normal controls [172] and liver, GEO Accession No. GSE115469, from 5 healthy patients [173]. The heart data are available on the Heart Cell Atlas database (https://www.heartcellatlas.org/) [174], 1 region was selected for download: the heart vascular datasets from 14 healthy donors. The brain data were searched at the scREAD database (https://bmblx.bmi.osumc.edu/scread/, accessed on 30 October 2024) [175], where the entorhinal cortex, GEO Accession No. GSE138852, from 4 healthy individuals [176]; the superior frontal gyrus, GEO Accession No. GSE147528, from 10 healthy individuals [177]; the prefrontal cortex, GEO Accession No. GSE129308, from 2 healthy controls [178], and the superior parietal lobe, GEO Accession No. GSE146639, from 5 non-demented elderly controls [179], were selected to constitute the organ dataset.

The data were loaded into R 4.1.0 and were preprocessed using standard parameters of the R packages ‘Seurat’ v.4.0.3 [180]. The expression matrix for each dataset was merged into one Seurat object using the “CreateSeuratObject” function. At the same time, cells with less than 100 expressed genes and higher than 25% mitochondrial genome transcript and genes expressed in less than three cells were removed. The gene expression data were normalized using the “NormalizeData” function, and the sources of cell–cell variation driven by batch were reverted out, using the number of detected UMI and mitochondrial gene expression by the ‘‘ScaleData’’ function. Highly variable genes were identified by the “FindVariableGenes” function and used for the principal component analysis (PCA) on the highly variant genes using the ‘‘RunPCA’’ function. The “JackStraw” function was implemented to remove the signal-to-noise ratio. Cells were then clustered utilizing the ‘‘FindClusters’’ function by embedding cells into a graph structure in PCA space. The clustered cells were then projected onto a two-dimensional space using the “RunUMAP” function. To create merged datasets of different organs, the “MergeSeurat” function was applied; the raw count matrices of two Seurat objects or more were merged into one, and a new Seurat object was created with the resulting combined raw count matrix.

## 4. Conclusions

In this study, the antileishmanial activity was evaluated of malvidin and echioidinin against *L. amazonensis*, *L. braziliensis*, and *L. infantum*. Malvidin demonstrated significantly higher in vitro efficacy, with lower IC50 values than echioidinin across both promastigote and axenic amastigote stages. Malvidin’s lower IC_50_ values compared to echioidinin further emphasize its potent antileishmanial properties. For instance, malvidin’s IC_50_ values for *L. amazonensis* were 197.71 ± 17 µM for stationary promastigotes and 258.07 ± 17 µM for axenic amastigotes, highlighting its potent antileishmanial properties

Cytotoxicity assays revealed that malvidin has a CC_50_ value of 2920.31 ± 80 µM, much higher than AmpB’s CC_50_ value of 1.06 ± 0.12 µM, indicating lower toxicity to mammalian cells. Favorable selectivity index (SI) values and minimal hemolytic activity further support malvidin’s therapeutic potential. Malvidin’s effectiveness was further demonstrated in infected macrophages, where it reduced infection rates in a concentration-dependent manner, lowering *L. amazonensis* infection rate to 59.30% at 120.74 µM.

The promising malvidin in vitro results enabled further in silico analysis that predicted favorable interactions between malvidin and the ARG enzyme, particularly with critical residues such as HIS139 and PRO258. These interactions, with binding energy values of −12.82 kcal/mol at a pH of 7 for *L. amazonensis*, suggest a plausible mechanism for its antileishmanial activity. Malvidin’s high electrophilicity and low global hardness enhance its adaptability within the ARG active site, reinforcing its ability to inhibit arginase and disrupt key metabolic pathways essential for parasite survival.

In this study, a detailed tissue-specific gene expression analysis was also conducted using single-cell RNA sequencing and network analysis. UMAP visualizations showed distinct brain, heart, kidney, and liver tissue clustering based on their gene expression profiles. Heatmap analysis revealed that malvidin treatment increased the expression of specific genes such as AXL in the heart and MAOA in the kidney, suggesting potential tissue-specific effects and benefits. Additionally, GLO1 and APEX1 were notably expressed across the heart, kidney, and liver, indicating malvidin’s role in enhancing cellular health by reducing oxidative stress and improving DNA repair mechanisms.

The combined in vitro, in silico, and gene expression results highlight malvidin’s potential as a low-toxicity antileishmanial candidate with specific ARG enzyme interactions. Its favorable selectivity index and macrophage infection rate reduction demonstrate therapeutic promise. Further in vivo studies and structure–activity relationship (SAR) explorations are needed to optimize its efficacy and fully develop its potential as a treatment for leishmaniasis.

## Figures and Tables

**Figure 1 molecules-30-00173-f001:**
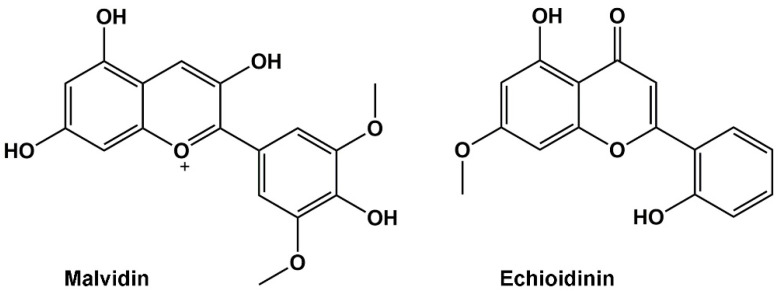
Chemical structures of malvidin (**left**) and echioidinin (**right**).

**Figure 2 molecules-30-00173-f002:**
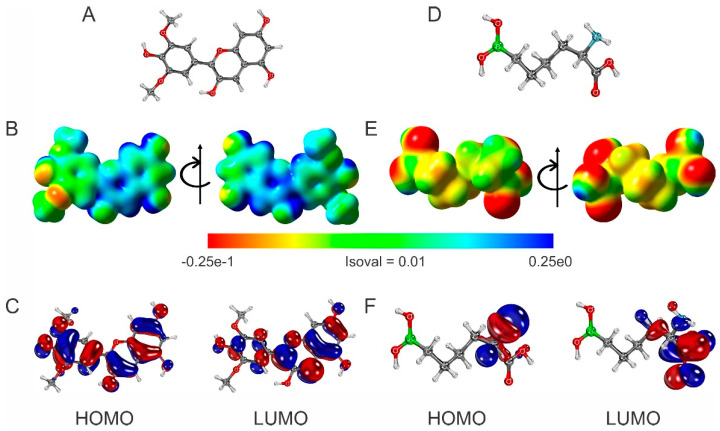
Analysis of ESP and frontier orbitals of malvidin and 2(S)-amino-6-boronohexanoic acid (ABH) compounds. (**A**) Optimized structure of malvidin. (**B**) ESP of malvidin. The range of colors corresponds to that described above. (**C**) HOMO and LUMO of malvidin. (**D**) Optimized structure of ABH. (**E**) ESP of ABH. The range of colors corresponds to that described above. (**F**) HOMO and LUMO of ABH.

**Figure 3 molecules-30-00173-f003:**
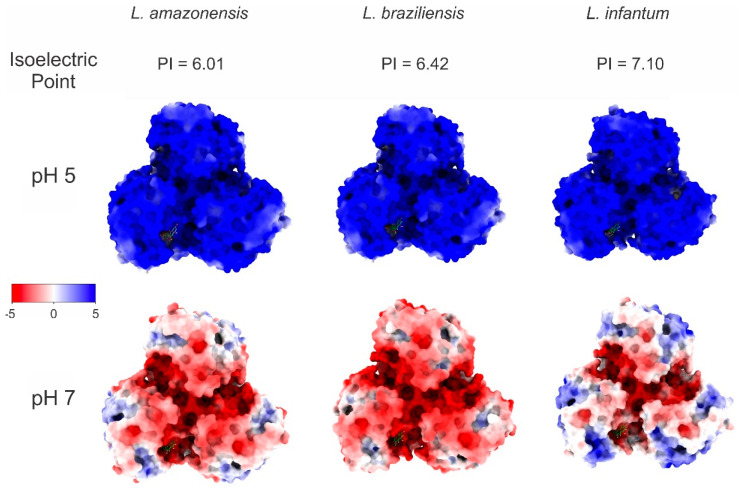
Representation of the electrostatic potential map of arginases (ARGs) for *L. amazonensis* (**left**), *L. braziliensis* (**middle**), and *L. infantum* (**right**) at a pH of 5 and a pH of 7 (in kBT/e). The electrostatic potential maps were calculated using the adaptive Poisson–Boltzmann solver (APBS) [63]. Blue represents positive potential, red represents negative potential, and white indicates neutral regions. Theoretical isoelectric point values were determined by ProtParam (https://web.expasy.org/protparam/ accessed on 15 October 2024), as part of the Expasy server [64].

**Figure 4 molecules-30-00173-f004:**
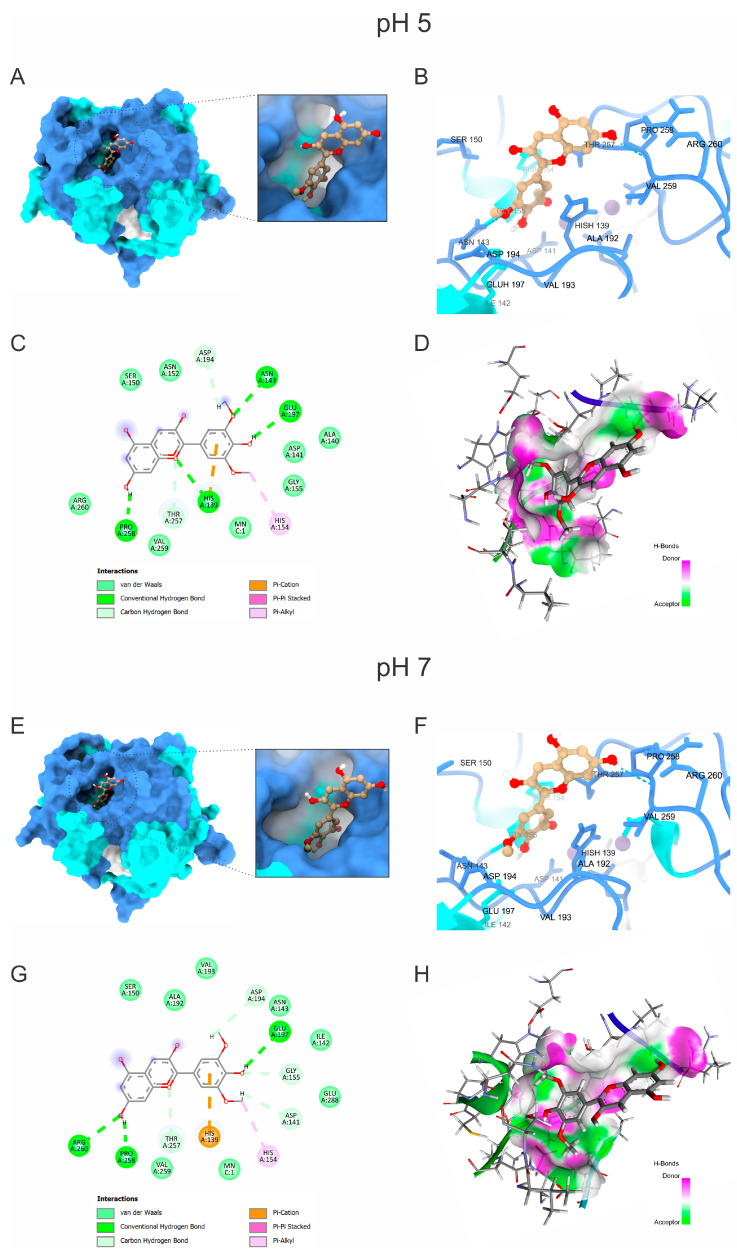
Docking of malvidin with arginase (ARG) from *Leishmania infantum* at a pH of 5. Panels (**A**–**D**) depict the 3-dimensional surface representation, the binding mode and molecular interactions of the interacting ligands, 2-dimensional view of ligand interactions with arginase residues, and hydrogen bonding interactions surface, respectively. Panels (**E**–**H**) reveal the corresponding docking analyses at a pH of 7.

**Figure 5 molecules-30-00173-f005:**
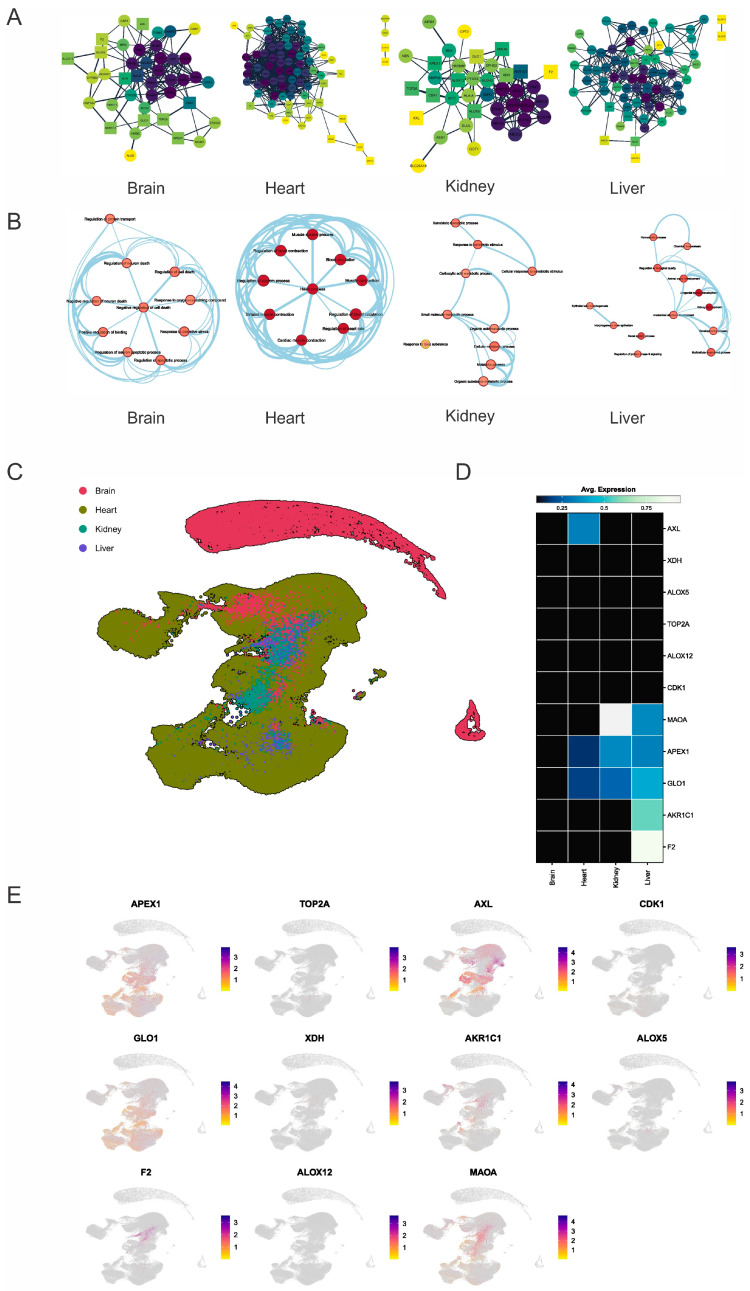
Single-cell RNA sequencing and comprehensive gene expression analysis across the brain, heart, kidney, and liver. (**A**) Network representations of gene interactions in the brain, heart, kidney, and liver, highlighting distinct patterns of connectivity and interaction hubs in each organ. Darker color represents maximal clique centrality (MCC). (**B**) Pathway analysis illustrating the major biological processes and pathways enriched in each organ, with significant nodes and connections visualized. (**C**) UMAP plot showing the clustering of gene expression data from the brain, heart, kidney, and liver, with each color representing a different organ. (**D**) Heatmap of average expression levels for key genes across the brain, heart, kidney, and liver, indicating organ-specific expression patterns. (**E**) Spatial expression maps for selected genes (APEX1, TOP2A, AXL, GLO1, XDH, AKR1C1, ALOX5, F2, ALOX12, MAOA, CDK1) across the different organs, with color intensity representing expression levels.

**Table 1 molecules-30-00173-t001:** In vitro antileishmanial activity IC_50_, selectivity index (SI), cytotoxicity (CC_50_), and hemolytic activity (RBC_50_) of malvidin, echioidinin, and pure amphotericin B (AmpB). The inhibitory concentrations at 50% observed for *Leishmania* spp. (IC_50_) and macrophages (CC_50_) were calculated using a sigmoidal regression to the dose–response curve. The results are presented as the mean ± standard deviation.

Compounds	IC_50_ (µM)			Selectivity Index (SI)			CC_50_ (µM)	RBC_50_ (µM)
	Stationary Promastigotes			Stationary Promastigotes				
	*L. amazonensis*	*L. braziliensis*	*L. infantum*	*L. amazonensis*	*L. braziliensis*	*L. infantum*		
Malvidin	197.71 ± 17	164.5 ± 19	141.26 ± 26	14.70	17.60	20.70	2920.31 ± 80	3120.13 ± 107
Echioidinin	272.99 ± 29	251.18 ± 22	212.83 ± 16	10.80	12.10	14.00	3010.62 ± 114	3400.76 ± 78
AmpB	0.08 ± 0.02	0.09 ± 0.03	0.06 ± 0.01	12.60	11.00	14.70	1.06 ± 0.12	14.28 ± 1
	Axenic amastigotes			Axenic amastigotes				
	*L. amazonensis*	*L. braziliensis*	*L. infantum*	*L. amazonensis*	*L. braziliensis*	*L. infantum*		
Malvidin	258.07 ± 17	227.89 ± 1	173.86 ± 19	11.30	12.70	16.70		
Echioidinin	335.96 ± 19	293.04 ± 20	261.73 ± 17	8.90	10.30	11.60		
AmpB	0.09 ± 0.02	0.10 ± 0.03	0.08 ± 0.02	11.00	9.80	12.60		

IC_50_, the concentration needed to inhibit 50% of the parasites’ viability; hSI, selectivity index, calculated by the ratio between CC_50_ and IC_50_; fCC_50_, the concentration required to inhibit 50% of the macrophages’ viability; gRBC_50_, the concentration needed to lysis 50% of the O+ human red blood cells.

**Table 2 molecules-30-00173-t002:** Treatment of infected macrophages and infection inhibition. The analysis determined the percentage of infection and the number of recovered amastigotes per infected cell after counting 200 macrophages in triplicate.

Compounds	Concentration (µM)	Infection After Treatment (%)	Number of Amastigotes per Macrophage	Infectiveness Reduction (%)
*L. amazonensis*				
Malvidin	120.74	59.30	3.80	35.75
	60.37	61.10	2.80	33.80
	30.18	75.60	3.30	18.09
	Control	92.30	4.40	0.00
Echioidinin	140.72	63.40	6.30	31.31
	70.36	71.80	3.70	22.21
	35.18	80.20	5.40	13.11
	Control	92.30	4.40	-
AmpB	5.41	24.20	2.80	73.78
	2.71	34.30	2.60	62.84
	Control	92.30	4.40	-
*L. braziliensis*				
Malvidin	120.74	39.80	3.30	42.07
	60.37	48.70	5.40	29.11
	30.18	52.10	4.40	24.16
	Control	68.70	5.50	-
Echioidinin	140.72	43.20	3.70	37.12
	70.36	50.80	3.70	26.06
	35.18	56.50	6.40	17.76
	Control	68.70	5.50	-
AmpB	5.41	16.50	2.40	75.98
	2.71	24.40	2.60	64.48
	Control	68.70	5.50	-
*L. infantum*				
Malvidin	120.74	49.80	2.70	33.15
	60.37	55.60	3.70	25.37
	30.18	61.20	4.70	17.85
	Control	74.50	3.70	-
Echioidinin	140.72	50.70	4.20	31.95
	70.36	58.70	2.80	21.21
	35.18	65.50	4.80	12.08
	Control	74.50	3.70	-
AmpB	5.41	19.80	2.50	73.42
	2.71	28.70	3.00	61.48

**Table 3 molecules-30-00173-t003:** Global reactivity of malvidin and ABH.

Compound	HOMO (eV)	LUMO (eV)	∆E Gap	µ	η	*S*	χ	ω	µ→ *
Malvidin	−6.32	−3.47	2.84	−4.89	1.42	0.35	4.89	8.41	5.57
ABH	−7.00	−0.41	6.59	−3.71	3.30	0.15	3.71	2.08	1.69

* μ→ corresponds to dipole moment (Debye).

**Table 4 molecules-30-00173-t004:** Average root mean square deviation (RMSD), radius of gyration (RG), and solvent accessible surface area (SASA) values obtained from molecular dynamics simulations.

	*L. amazonensis*	*L. braziliensis*	*L. infantum*
RMSD (nm)			
pH 5	0.023 ± 0.001	0.023 ± 0.001	0.023 ± 0.001
pH 7	0.023 ± 0.001	0.023 ± 0.001	0.023 ± 0.001
RG (nm)			
pH 5	3.017 ± 0.001	3.017 ± 0.001	3.017 ± 0.001
pH 7	3.016 ± 0.001	3.017 ± 0.001	3.017 ± 0.001
SASA (nm^2^)			
pH 5	332.244 ± 1.530	344.397 ± 1.517	335.716 ± 1.504
pH 7	331.772 ± 1.494	344.434 ± 1.552	334.429 ± 1.430

**Table 5 molecules-30-00173-t005:** Binding free energy calculation of malvidin–ARG complexes using molecular mechanics generalized born surface area (MM/GBSA).

Energy *	*L. amazonensis*		*L. braziliensis*		*L. infantum*	
	pH 5	pH 7	pH 5	pH 7	pH 5	pH 7
ΔE_vDW_	−29.49	−28.82	−30.78	−34.12	−26.37	−26.96
ΔE_EL_	148.22	−90.93	134.83	−157.23	177.00	−22.25
ΔE_GB_	−124.69	110.87	−108.55	183.44	−159.78	42.08
ΔE_SURF_	−3.77	−3.94	−3.91	−4.14	−3.68	−3.72
ΔG_GAS_	118.72	−119.74	104.06	−191.35	150.63	−49.21
ΔG_SOLV_	−128.46	106.92	−112.47	179.30	−163.46	38.36
ΔG_TOTAL_	−9.73	−12.82	−8.41	−12.05	−12.83	−10.85

* All values are in kcal/mol.

**Table 6 molecules-30-00173-t006:** Global reactivity descriptors.

Reactivity Descriptors	Equation	References
Chemical Potential (µ)	$µ=\frac{1}{2}(\varepsilon_{L}+\varepsilon_{H})$	[153,154]
Global Hardness (η)	$\eta =\frac{1}{2}(\varepsilon_{L}-\varepsilon_{H})$	[153,154]
Global Softness (*S*)	$S=\frac{1}{\eta }$	[153]
Electronegativity (χ)	χ=−µ	[153]
Electrophilicity (ω)	$\omega =\frac{{µ}^{2}}{2\eta }$	[153,155]
HOMO–LUMO gap (∆E gap)	$∆Egap=\varepsilon_{L}-\varepsilon_{H}$	

## Data Availability

The software used in this study includes GROMACS and VMD v. 1.9.4, which are available to non-commercial users under specific distribution licenses, and CHARMM-GUI, which is free for academic use and requires a paid license for commercial purposes. All software tools are referenced throughout the Methods and Results sections. Additionally, structures, trajectories, and analysis scripts generated during this work are documented accordingly. All trajectories, structures, simulation scripts, analysis scripts, and data files will be shared with this work through a link hosted on the Amaro Lab Website (https://amarolab.ucsd.edu/covid19.php).

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
