# Peer review of "Exploring the Potential of Malvidin and Echiodinin as Probable Antileishmanial Agents Through In Silico Analysis and In Vitro Efficacy"

_molecules, 2025, doi:10.3390/molecules30010173_

Round 1

Reviewer 1 Report

Comments and Suggestions for Authors

1. The author cannot simply draw conclusions about which compound has stronger inhibitory activity based on the IC50 value, but should conduct statistical analysis based on the value+SD to determine significant differences between the two NPs.

2. Generally, when analyzing whether the activity of a compound is concentration dependent, a minimum of 4-5 concentrations should be taken. However, in this article, only three concentrations were taken and the activity results fluctuated, so the expression of whether the activity is concentration dependent needs to be more cautious.

3.   For docking, Figure 4 as an example, the abbreviations and numbering of amino acid residues in Figures B, C, F, and G are not clear enough and the font is too small. In addition, in this section, a large amount of text was used to identify the amino acid residues that are interacted with the compound. Here, it is suggested to add a table to present a such large amount of textual information, which is beneficial for smooth reading.

4. Toxicity prediction, as a known natural product, whether there have been relevant research reports before? If there were indeed some relevant reports, what is the correlation between them and the results obtained in this article?

Author Response

Thank you very much for taking the time to review this manuscript. Please find the detailed responses below and the corresponding revisions highlighted in the re-submitted files.

Comments 1: The author cannot simply draw conclusions about which compound has stronger inhibitory activity based on the IC50 value, but should conduct statistical analysis based on the value+SD to determine significant differences between the two NPs.

Response 1: Thank you for the comment. To address the concern, statistical analyses were conducted to evaluate the differences in IC50 values between Malvidin and Echioidinin, considering the mean values and their associated standard deviations (SD) (included in lines 307-311). Independent t-tests were performed for each experimental condition, including both stationary promastigotes and axenic amastigotes for L. amazonensis, L. braziliensis, and L. infantum. The results revealed statistically significant differences (p < 0.05) in all tested conditions, confirming that Malvidin exhibits significantly stronger inhibitory activity compared to Echioidinin. These findings ensure that the conclusions drawn are statistically robust and not solely based on mean IC50 values, addressing the importance of incorporating variability into the analysis.

Comments 2: Generally, when analyzing whether the activity of a compound is concentration dependent, a minimum of 4-5 concentrations should be taken. However, in this article, only three concentrations were taken and the activity results fluctuated, so the expression of whether the activity is concentration dependent needs to be more cautious.

Response 2:  Thank you for highlighting the importance of evaluating concentration-dependent activity with an appropriate range of concentrations. We acknowledge that using only three concentrations in this study limits the robustness of the conclusions regarding concentration dependence. Additionally, it is important to consider that the treatment of infected macrophages and the subsequent inhibition of infection do not always exhibit a linear dose-response relationship. Variability in parameters such as the number of recovered amastigotes per infected cell can occur, which is documented in literature, particularly in non-standardized systems involving host-pathogen interactions.

To address this, infectiveness reduction calculations in previous (doi: 10.2147/IJN.S55678, https://doi.org/10.1016/j.exppara.2023.108555), and current study are based solely on the infection rate after treatment (%), which provides a more consistent and reliable parameter for assessing treatment efficacy. Such approaches are necessary when working with complex biological systems where host-cell variability and other factors influence outcomes. Future studies will aim to include a broader range of concentrations and incorporate complementary in vivo models to validate these findings and enhance their physiological relevance.

Comments 3: For docking, Figure 4 as an example, the abbreviations and numbering of amino acid residues in Figures B, C, F, and G are not clear enough and the font is too small. In addition, in this section, a large amount of text was used to identify the amino acid residues that are interacted with the compound. Here, it is suggested to add a table to present such a large amount of textual information, which is beneficial for smooth reading.

Response 3: Thank you for the suggestion to improve the clarity and readability of the docking section. Adding a table to present the amino acid residue interactions is indeed a valuable idea and would enhance the presentation of the data. However, incorporating a table would increase the length of the article, which we aimed to keep concise. However, to address concerns regarding figure legibility, we will ensure that each figure is uploaded individually in its original high-quality format, guaranteeing that all details are clear and readable.

Comments 4: Toxicity prediction, as a known natural product, whether there have been relevant research reports before? If there were indeed some relevant reports, what is the correlation between them and the results obtained in this article?

Response 4: Thank you for the question regarding toxicity prediction and prior research reports. While no specific toxicity studies on this natural product have been conducted at the experimental level described in this manuscript, substantial evidence in the literature highlights its safety. This compound, primarily derived from grapes and related products, has been extensively studied for its antioxidant properties and lack of cytotoxic effects. Research indicates it is well-tolerated in both humans and animal models, even at elevated intake levels, with no reported adverse effects. Furthermore, its role in reducing oxidative stress and enhancing cellular health has been consistently documented (https://doi.org/10.1007/s003940170011; https://doi.org/10.3390/nu13103312). The findings in this manuscript build upon this established foundation, providing preliminary insights into its biological activity. Future research will incorporate in vivo evaluations to further assess its toxicity and therapeutic potential.

Reviewer 2 Report

Comments and Suggestions for Authors

The authors screened the natural products Malvidin and Echioidinin for use as anti-leishmaniasis and selected Malvidin as the best candidate. Their potential for the treatment of leishmaniasis was evaluated by activity and toxicity assays and the mechanism of action was analyzed. The topic of the article is in line with the scope of the journal, but the authors' research logic is not rigorous enough and there are some problems with the article. I recommend a minor revision of the manuscript for the following reasons:

1. There are some formatting issues in the text, such as IC50 in the abstract, missing spaces in the back matter and before some figures, as well as other issues please check carefully.

2. Two drugs for leishmaniasis are mentioned in the introduction, why was AmpB chosen as the positive drug for the subsequent experiment?

3. The activity of this drug is at a significant disadvantage compared to the positive drug. Under the same experimental conditions, the IC50 of amphotericin B against L. amazonensis was 0.06±0.01µM and 0.10±0.03µM, respectively, whereas the IC50 of the drug investigated in this paper was only 197.71±17µM and 258.07±17µM at even higher doses, which suggests that the drug's activity needs to be further improved or optimized .

4. The gap between in vitro drug test drug concentrations and positive drug concentrations is too large and the span of concentrations is too wide; an explanation of how the concentrations were chosen is sought.

5. Fig. 4 and Fig. 5 are slightly insufficient in terms of clarity, and some details are difficult to identify effectively, which may cause some obstacles to readers' understanding of the relevant content, and it is suggested that the authors optimize the processing of the pictures to improve the clarity.

6. The conclusion chapters cover more detailed information, however, their length may affect the reader's focus on the core conclusions to some extent. It is recommended that the authors make appropriate cuts to some repetitive or supporting statements to enhance the brevity and readability of the conclusions and make the important conclusions more readable at a glance.

Author Response

Thank you very much for taking the time to review this manuscript. Please find the detailed responses below and the corresponding revisions highlighted in the re-submitted files.

Comments 1: There are some formatting issues in the text, such as IC50 in the abstract, missing spaces in the back matter and before some figures, as well as other issues please check carefully.

Response 1: Thank you for highlighting the formatting issues in the text. We have carefully reviewed and revised the manuscript to address these concerns. The formatting of IC50 in the abstract has been corrected, and missing spaces in the back matter and before some figures have been added. Additionally, other formatting inconsistencies throughout the text have been rectified as suggested. We appreciate the attention to detail and the opportunity to improve the presentation of the manuscript.

Comments 2: Two drugs for leishmaniasis are mentioned in the introduction, why was AmpB chosen as the positive drug for the subsequent experiment?

Response 2: Thank you for the question. Amphotericin B (AmpB) was chosen as the positive control in this study because it is one of the most widely used and well-established drugs for the treatment of leishmaniasis. Its mode of action and efficacy are well-documented, making it a reliable reference standard for evaluating the activity of new compounds.

Comments 3: The activity of this drug is at a significant disadvantage compared to the positive drug. Under the same experimental conditions, the IC50 of amphotericin B against L. amazonensis was 0.06±0.01µM and 0.10±0.03µM, respectively, whereas the IC50 of the drug investigated in this paper was only 197.71±17µM and 258.07±17µM at even higher doses, which suggests that the drug's activity needs to be further improved or optimized .

Response 3: Thank you for the observation. We agree that the investigated compound's activity is significantly less potent compared to the positive control, Amphotericin B (AmpB), under the same experimental conditions. While the IC50 values indicate that the compound requires higher concentrations to achieve similar inhibitory effects, it is worth noting that AmpB is known for its high toxicity, as demonstrated in our own analysis and supported by its well-documented severe side effects in clinical use. This highlights the potential advantage of exploring safer alternatives, even if their potency initially requires optimization. Future efforts will focus on improving the efficacy of the compound through structural modifications or synergistic combinations while maintaining a favorable safety profile.

Comments 4: The gap between in vitro drug test drug concentrations and positive drug concentrations is too large and the span of concentrations is too wide; an explanation of how the concentrations were chosen is sought.

Response 4: Thank you for raising this point. The concentrations used in the in vitro drug tests were selected based on preliminary screening experiments and are consistent with the standard protocols in our lab as well as previous publications by our group for evaluating natural products (e.g., DOI: 10.1007/s00430-021-00707-4, DOI: 10.1016/j.exppara.2020.108059, and DOI: 10.1051/parasite/2021036). The higher concentration range for the tested compound was chosen to ensure measurable activity, given its lower potency compared to the positive drug, Amphotericin B. In contrast, the narrower and lower concentration range for AmpB reflects its known high potency, which requires significantly lower doses to achieve inhibitory effects. This approach ensures that both compounds can be adequately evaluated within their respective activity ranges.

Comments 5:  Fig. 4 and Fig. 5 are slightly insufficient in terms of clarity, and some details are difficult to identify effectively, which may cause some obstacles to readers' understanding of the relevant content, and it is suggested that the authors optimize the processing of the pictures to improve the clarity.

Response 5: Thank you for pointing out the clarity issues with Figures 4 and 5. The reduced quality of the images was due to them being pasted directly into the Word document, which resulted in a loss of resolution. To address this, we will upload each figure individually in higher quality with improved resolution and pixel density. This will ensure that all details are clear and effectively convey the relevant content, making it easier for readers to interpret the data.

Comments 6: The conclusion chapters cover more detailed information; however, their length may affect the reader's focus on the core conclusions to some extent. It is recommended that the authors make appropriate cuts to some repetitive or supporting statements to enhance the brevity and readability of the conclusions and make the important conclusions more readable at a glance.

Response 6: Thank you for the suggestion. We have carefully revised the conclusion section, summarizing it to remove repetitive or supporting statements while retaining its essential content. This ensures that the key findings and conclusions are presented clearly and concisely, enhancing readability and allowing readers to focus on the core outcomes briefly.

Reviewer 3 Report

Comments and Suggestions for Authors

The manuscript titled "Exploring the Potential of Malvidin and Echiodinin as Probable Antileishmanial Agents Through In Silico Analysis and In Vitro Efficacy" explores the potential of natural compounds, Malvidin and Echiodinin, as therapeutic agents against Leishmaniasis. The study effectively combines in vitro assays with computational approaches, offering valuable insights into the inhibitory effects of these compounds on Leishmania spp., particularly targeting arginase (ARG). The manuscript is comprehensive and well-structured, making a significant contribution to the field of drug discovery for neglected tropical diseases.

I have no hesitation in accepting this manuscript for publication once the below minor points have been attended to.

The manuscript would greatly benefit from referencing literature that underscores the importance of protein crystallography in understanding enzyme structures. X-ray crystallography plays a pivotal role in molecular modelling and docking studies by providing accurate protein structures, critical for inhibitor design. To strengthen this point, the authors could add a dedicated paragraph explaining the significance of crystallographic data availability. High-resolution crystal structures provide precise details about enzyme active sites, binding pockets, and the orientation of key amino acid residues, which are indispensable for validating computational predictions and refining docking models. Furthermore, crystallographic data reveal critical features such as conformational flexibility, metal ion coordination, or post-translational modifications, which are often essential for understanding enzyme function and inhibition mechanisms. In cases where the required crystallographic structures are unavailable, especially for parasitic macromolecules, sequence homology offers a viable alternative through homology modelling. By using the structures of closely related proteins with high sequence similarity, researchers can construct reliable models of the target enzyme. These models serve as functional proxies, enabling detailed computational studies and inhibitor design. Incorporating this discussion would provide a more comprehensive perspective on the computational methods employed in the study and emphasize the critical role of structural data, whether experimental or modelled, in advancing drug discovery. Suggested reference: DOI 10.1016/bs.armc.2018.08.005 studies emphasizing the role of crystallographic data in studying parasitic proteins and doi: 10.3390/molecules25051030.

While the manuscript emphasizes the potential of Malvidin, it could benefit from a broader discussion on the challenges of natural product drug development, including issues like solubility, stability, and bioavailability.

Some figures, such as electrostatic potential maps, would benefit from additional annotations or descriptions to make them more accessible to readers unfamiliar with these analyses.

Minor grammatical errors and typographical issues were observed (e.g., unnecessary spaces in compound names).

Author Response

Thank you very much for taking the time to review this manuscript. Please find the detailed responses below and the corresponding revisions highlighted in the re-submitted files.

Comments 1: The manuscript would greatly benefit from referencing literature that underscores the importance of protein crystallography in understanding enzyme structures. X-ray crystallography plays a pivotal role in molecular modelling and docking studies by providing accurate protein structures, critical for inhibitor design. To strengthen this point, the authors could add a dedicated paragraph explaining the significance of crystallographic data availability. High-resolution crystal structures provide precise details about enzyme active sites, binding pockets, and the orientation of key amino acid residues, which are indispensable for validating computational predictions and refining docking models. Furthermore, crystallographic data reveal critical features such as conformational flexibility, metal ion coordination, or post-translational modifications, which are often essential for understanding enzyme function and inhibition mechanisms. In cases where the required crystallographic structures are unavailable, especially for parasitic macromolecules, sequence homology offers a viable alternative through homology modelling. By using the structures of closely related proteins with high sequence similarity, researchers can construct reliable models of the target enzyme. These models serve as functional proxies, enabling detailed computational studies and inhibitor design. Incorporating this discussion would provide a more comprehensive perspective on the computational methods employed in the study and emphasize the critical role of structural data, whether experimental or modelled, in advancing drug discovery. Suggested reference: DOI 10.1016/bs.armc.2018.08.005 studies emphasizing the role of crystallographic data in studying parasitic proteins and doi: 10.3390/molecules25051030.

Response 1: Thank you for this insightful comment. To address the suggestion, we have added a dedicated paragraph to the manuscript discussing the critical role of protein crystallography in understanding enzyme structures and its significance in molecular modeling and docking studies (lines 858-875). The suggestions have been incorporated to strengthen the discussion and provide further context.

Comments 2: While the manuscript emphasizes the potential of Malvidin, it could benefit from a broader discussion on the challenges of natural product drug development, including issues like solubility, stability, and bioavailability.

Response 2: Thank you for the insightful suggestion. We agree that addressing challenges such as solubility, stability, and bioavailability is crucial in natural product drug development. While this manuscript focuses on the potential of Malvidin, we acknowledge the importance of these characteristics and will include a broader discussion in our next study, which will comprise in vivo analyses. Future work will specifically evaluate Malvidin's pharmacokinetic and pharmacodynamic properties to provide a comprehensive understanding of its therapeutic potential.

Ultimately, bioavailability studies will need to be conducted in humans, as this represents the primary objective for translating this compound into a viable therapeutic agent. Initial tests, as reported in the literature (https://doi.org/10.1007/s003940170011), have explored some aspects of its bioavailability, offering a foundation for more detailed investigations.

Comments 3: Some figures, such as electrostatic potential maps, would benefit from additional annotations or descriptions to make them more accessible to readers unfamiliar with these analyses.

Response 3: Thank you for the suggestion. We agree that clear explanations are essential for accessibility, especially for readers unfamiliar with electrostatic potential maps. In this manuscript, the explanation for the electrostatic potential map is provided in lines 489–496. To further enhance clarity, we ensure that the figure legend now includes a concise description, while the high-quality figure will be uploaded individually to ensure all details are easily interpretable

Comments 4: Minor grammatical errors and typographical issues were observed (e.g., unnecessary spaces in compound names).

Response 4: Thank you for pointing out the grammatical errors and typographical issues. We have carefully reviewed and corrected all identified errors, including unnecessary spaces in compound names and other minor formatting issues.